# Factors Influencing Entrepreneurial Intention: Focusing on Individuals' Knowledge Exploration and Exploitation Activities

**Saerom Lee [1], Min-Jeong Kang [2] and Byung-Keun Kim [3,]***

1    School of Business Administration, Kyungpook National University, Daegu 41566, Korea
2    BK21 Business Education Research Center, Korea University, Seoul 02841, Korea
3    School of Industrial Management, Korea University of Technology and Education, Cheonan 31253, Korea
*    Correspondence: b.kim@koreatech.ac.kr

**Abstract:** With rapid changes in industrial environments, the role of start-ups and their initial attempts toward the market are regarded as critical initiates. To increase the number of young start-up entrepreneurs, it is important to understand the relationship between entrepreneurial intention and its antecedents. Drawing on the organizational learning theory, we adopt the concept of exploration and exploitation as mediating roles of entrepreneurial intention as individual-level perspectives. Additionally, we consider innovativeness, self-efficacy, and internal locus of control as antecedents of two learning activities. The results indicate that each learning activity successfully supports the entrepreneurial intention, which requires a complex decision-making process and long-term efforts. This research discusses practical implications and guidelines for entrepreneurship education programs further.

**Keywords:** entrepreneurial intention; young start-up entrepreneur; organizational learning; exploration; exploitation; innovativeness; self-efficacy; internal locus of control





## 1. Introduction

Multiple nations have begun to focus on innovation growth through start-ups as a primary strategy for economic development [1]. Young start-up entrepreneurs are encouraged to develop new employment opportunities to resolve unemployment among young people led by various factors, including slowdowns in economic growth. The term "start-up" refers to the establishment of an innovative process or system that produces and sells goods or services as a business [2]. The establishment of a new enterprise requires that entrepreneurs have the ability to elicit ideas from the surrounding environment and capture business opportunities [2]. Since start-ups require a long-term process of planning and implementation, establishing a new business depends on entrepreneurs' fixed temperament as well as personal motivations and attitudes. Most previous research focuses on individual characteristics as antecedents of entrepreneurial intentions, which are necessary conditions for starting a new business [3]. Considering that the complexities and problem-solving processes required for starting a new business demand knowledge of various fields, this study explains individual learning behaviors based on the organizational learning theory to explore and use the knowledge and experience needed for successful start-ups. In this study, we assume that the relationship between personal traits and entrepreneurship intention is not directly associated with, but the relationship can be mediated by the ability to recognize various environments and situations and obtain and utilize information appropriate to the situation. This research tries to examine the role of learning activities. The study also explores the characteristics that affect individuals' exploration and exploitation activities, presenting a fundamental approach and a systematic direction that delineates the individual capabilities needed to increase start-up leaders' entrepreneurial intentions.

Existing research suggests that learning about and direct and indirect experience with participating in starting a start-up can affect start-up initiatives [4]. To specify the learning

behaviors needed from entrepreneurs, organizational learning theory is adopted to explain organizational learning activities for advancing firms' innovative performance. Organizational learning theory specifies the process of searching for, identifying, and exploring knowledge that is external to the company, called exploration activities, and the creation of new knowledge using the inherent knowledge within a company refers to exploitation activities [5]. In previous research, the concepts of exploration and exploitation have been applied to organizational-level research and measured by investment in R&D activities or patenting characteristics [6–8]. In this vein, few studies have examined individual exploration and exploitation activities. Recently, a few studies started to emphasize that individuals also could engage in the routine of learning activities, which results in organizational performance [9,10]. For example, Ref. [10] focus is given to individual exploration and exploitation activities for incentive-given performance, asserting that without clear incentives, individuals lack motivation for exploration and exploration activities. Initiating start-ups requires various types of experience and knowledge in the long term, and the ability to search for and use knowledge is directly linked to entrepreneurs' ability to navigate when problems arise or when new knowledge is needed. Therefore, this study suggests that learning activities have a direct influence on entrepreneurial intentions.

For start-ups, the process of successfully starting a business and earning a profit is arduous and faces a variety of challenges that require solutions, so the founder's individual characteristics will have an influence on the organizational learning process when accumulating a variety of knowledge and experience. To reveal the link between entrepreneurs' personal characteristics and entrepreneurial intentions, we investigate research questions regarding learning activities: How do exploration and exploitation activities mediate personal traits and entrepreneurial intentions? Personal traits are mainly included in previous studies that engaged planned behavior theory, adopted as antecedents that can affect exploration and exploitation activities. This study focuses on innovativeness, self-efficacy, and internal locus of control as personal traits.

This study's theoretical contributions are threefold. First, this study reveals the mediating variables between personal traits and entrepreneurial intentions. Most previous research indicates that entrepreneurs' traits and intention to engage in start-up education activities are directly related to entrepreneurial intentions [11]. However, not every individual with these identified personal traits will become an entrepreneur. Because the start-up itself faces various difficulties and challenges during the foundational process, entrepreneurs require new knowledge and experience to meet and solve arising problems. Thus, to successfully start a business, entrepreneurs' learning activity, which is a more direct endeavor, will play a major role in understanding the origin and motivations of entrepreneurial intentions. Second, this study measures exploration and exploitation from an individual-level perspective. Previous research in the field of strategic management has investigated organizational learning theory from a firm-level perspective [12,13]. Specifically, most research measures exploration and exploitation through proxies such as patent, R&D budget expenditure, or mergers and acquisitions (M&A) [6–8]. Considering that exploration and exploitation are important individual activities when people gain knowledge, there remains relatively limited research regarding learning activities at the individual level. Third, this study identifies personal traits that act as antecedents of engaging in learning activities. There is a lack of research regarding how specific personal traits activate individual attitudes toward founding start-ups and motivate founders to prepare for long-term learning activities. The mediating variables of learning activities that represent core capabilities for overcoming challenges and procuring new knowledge when managing a start-up are categorized into exploration and exploitation. In addition, this study specifically examines how personal traits influence the two activities. To the best of our knowledge, no research has examined the antecedents of individuals' proclivity toward exploration and exploitation activities.

## 2. Literature Review

### 2.1. Entrepreneurial Intention

Entrepreneurial intention refers to an individual's objective to start a high-growth business and work as an entrepreneur in the future [14]. Ajzen & Fishbein [15] refers to entrepreneurial intentions as an act or intentional attitude toward a start-up or an existing firm attempting to start a new business or seeking new opportunities. Kuratko et al. [16] assert that entrepreneurial intention refers to the development of a practical and specific plan for starting a business. It Is also interpreted as a desire to own and start a private business, including plans to become entrepreneurs [17]. Because entrepreneurial intention is considered to be a precursor to the act of starting a business [1], it is a prerequisite mindset for initiating a start-up. Entrepreneurial intention is embodied in an entrepreneur's personal efforts to implement start-up activities, applying individual interest and actions to starting a business [17]. Entrepreneurs' engagement with the possibility of starting their own businesses in the future is also considered a positive attitude toward start-ups [18,19]. Entrepreneurial intention must necessarily precede founding a start-up. Katz and Gartner [20] regarded individuals' entrepreneurial intention as an important factor for predicting entrepreneurship, applying the theory of planned behavior.

Various studies have analyzed the antecedents that influence entrepreneurial intention. Bird [21] divided the factors affecting entrepreneurial intention into entrepreneurs' personal characteristics (e.g., personality, ability, and experience) and surrounding environmental factors. Individual entrepreneurial characteristics are exerted and learned through interactions with the environment and related entrepreneurship activities [22]. The major individual-level antecedents include satisfaction with entrepreneurship education [23], entrepreneurship education [24], motivation to participate in entrepreneurship education [25], and attitude and experience [26]. In addition, existing studies have focused on founders' psychological tendencies and the personality factors that influence entrepreneurial intention. Turker and Selcuk [27] examined the important personality traits that affect entrepreneurial intention, such as self-confidence, risk-taking ability, need to achieve, and locus of control.

Covin and Slevin [28] tested innovativeness, risk-taking, and proactiveness as antecedents of entrepreneurship. In [29], an additional analysis was performed considering the founder's personal characteristics, environmental characteristics, and entrepreneurship-related characteristics. Specifically, achievement needs and self-efficacy as individual characteristics; the existence of an entrepreneurship mentor as environmental characteristics; and social awareness and business strategy, and risk sensitivity and innovation as entrepreneurship characteristics were verified. In addition, [30] and [31] specifically analyzed the effect of learning orientation, activity orientation, and goal orientation on entrepreneurial intention. Wong and Choo [32] also indicated that entrepreneurial intention requires the ability to search for and obtain information that can be leveraged to accomplish the goals of venture creation. The ability to procure vital information and knowledge to overcome challenges that arise during the start-up process is a crucial element to entrepreneurs. Because previous research on entrepreneurial intention was unable to identify the different consequences of similar personal traits, this study endeavors to reveal the hidden link between personal traits and entrepreneurial intention by focusing on individual learning abilities.

Recently, entrepreneurial intention-related research has started to investigate the factors that mediate personal traits and entrepreneurial intention. With the same vein, we tried to find factors as mediators. For example, Ref. [33] examined how entrepreneurial alertness mediates when big five personality traits influence entrepreneurial intention. Gill et al. [34] also empirically investigated whether entrepreneurial self-efficacy mediates the influence when cognitive flexibility and entrepreneurial awareness affect entrepreneurial intention. In addition, Ref. [35] examined the mediating role of attitude and creative-process engagement in the relationship between creative self-efficacy and entrepreneurial intention. Although many recent studies have only looked at the direct path of personal traits that

affect entrepreneurial intention, recently, even if they have the same personal trait, the studies have started to focus on mediators that are unique abilities of entrepreneurs or prospective entrepreneurs.

*2.2. Exploration and Exploitation Activities*

Organizational learning theory is defined as the experiential production and reproduction of organizational rules and processes that lead to behavioral stability or behavioral change in organizations [12,13]. Organizational learning theory has been treated as a strictly organizational characteristic in the strategy field, examining the ways in which companies share knowledge or communicate within a company and becoming an organizational rule for accepting and creating new information or knowledge. According to [5], organizational learning theory is divided into two aspects: exploration and exploitation. Exploration refers to learning activities to research and gain knowledge that is not within a company, from outside of the company to advance innovative outcomes. Examples of exploration activities include investment in R&D for innovative product development, hiring employees with needed skills that are unfamiliar to the company, M&A with a different technology-based company, or formally acquiring knowledge from a university or other education center. In contrast, exploitation refers to developing new knowledge or technology through leveraging the existing knowledge possessed within the company. Launching an expanded or functionally improved form of a product already being made is closely related to exploitation activity. Product improvement and knowledge generation are often motivated by reflecting on customers' detailed complaints, suggestions, or improvement requirements. M&A between companies with similar technologies is also considered a form of exploitation.

The delineations of exploration and exploitation as organizational learning activities have been adapted to examine individuals' learning activities [9,36,37]. Mom, Van Den Bosch, and Volberda [38] defined managers' explorative behavior as "searching for, discovering, creating, and experimenting with new opportunities" and exploitative behavior as "selecting, implementing, improving, and refining existing certainties" (p. 910). Although it is not possible to measure organizational learning methods at an individual level in the same way [39], many studies emphasize that individuals can also engage in the two activities. It is said to have a positive effect on firms' performance in practice [9,36,38,40,41]. Few studies have examined why individuals who engage in both activities are motivated and how this affects organizations. Lee and Meyer-Doyle [10] studied the effect of employee exploration and exploitation activities using performance-based incentives. Their research focuses on the incentive for each activity as individuals' motivation to participate. Mom et al. [42,43] and Good and Michel, [44] indicated that individual learning activities influence the high-performance work system. Rosing and Zacher [45] discuss the impact of balancing the learning behaviors of individuals performing both exploration and exploitation on innovation performance, adopting questionnaires to investigate individuals' exploration and exploitation activities in daily life. Recently, Ref. [46] analyzed how individuals' ambidextrous behaviors are being conducted by focusing on individual employees' knowledge seeking and offering activities through a survey.

Previous studies related to entrepreneurship have examined how entrepreneurship orientation is related to organizational learning within companies [47]. The number of studies has been limited to organizational-level studies, suggesting that the corporate atmosphere and environment have a positive effect on the organizational learning process at the firm level, which in turn has a positive effect on innovation performance. On the contrary, this study differs from previous studies in that it studies exploration and exploitation activities, which are individual activities that affect individual entrepreneurial intention. This study focuses on the effect of individual learning activities on entrepreneurial intentions on the long-term and plan-based activities of entrepreneurship. In a broader sense, this study examines the whole of the personal direct and indirect experiences and learning

activities that affect entrepreneurial intention, emphasizing the importance of the processes of exploring and exploiting knowledge.

## 3. Research Model

Figure 1 presents the research model of this study. The research model assumes that entrepreneurs' individual traits, such as innovativeness, self-efficacy, and internal locus of control, positively influence exploration and exploitation activities as mediators of entrepreneurial intention. In this study, a study was conducted for prospective entrepreneurs who should perform entrepreneurship based on innovative ideas. In this study, innovative output was assumed to be the starting start-up, and the research model was presented by limiting the research model to innovativeness, self-efficacy, and internal locus of control by focusing on individual characteristics that can affect the learning activities.

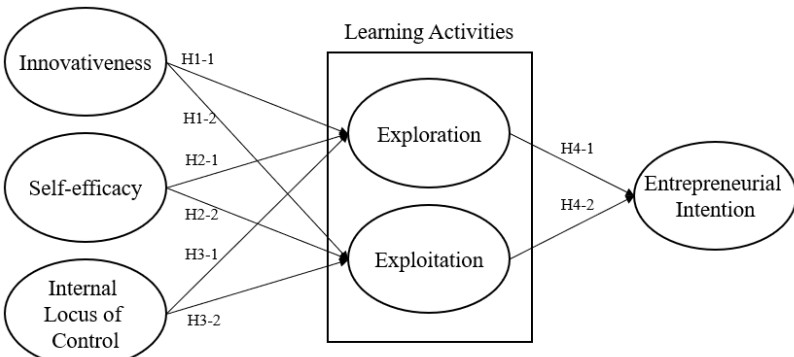

**Figure 1.** Research model.

## 4. Hypotheses

Innovation is the ability to pursue change and leverage innovative ideas to identify new opportunities and solve existing problems [48]. Many studies related to entrepreneurship assert that innovation is the main characteristic of entrepreneurs (e.g., [49,50]. Various studies investigate exploration and exploitation activities as learning activities of organizations that can potentially lead to corporate innovation performance. In contrast, this study seeks to examine the effect of individual innovation on exploration activities among learning activities. Although various research has attempted to reveal the link between individuals' behavior and organizational learning activities, most has focused on the firm-level performance as the dependent variable [39]. For instance, [51] examined the impacts of top managers' individual innovation behaviors on firm-level innovation activities, with a focus on CEO leadership. However, there remains a lack of research on individuals' innovative behavior with regard to exploration and exploitation activities compared to studies about the impacts of those activities on firms' overall innovation performance. This study endeavors to fill the gap of previous research by specifically focusing on innovation resulting from entrepreneurs' personal traits in terms of exploration activities. Continuous innovation is required in an environment of high uncertainty and for developing novel ideas. Therefore, this study targets exploration and exploitation activities to learn about new ideas during the whole process of venturing, which requires entrepreneurs' innovativeness.

**Hypothesis 1.1.** *Entrepreneurs' innovativeness positively influences exploration activities.*

**Hypothesis 1.2.** *Entrepreneurs' innovativeness positively influences exploitation activities.*

The role of self-efficacy is intensively studied among personal factors that influence knowledge-management activities, such as knowledge sharing, in previous studies. Self-efficacy is also used as a major individual variable for mediating or moderating environmental factors that affect individual performance. For example, [52] demonstrated the moderating role of self-efficacy when corporate environmental factors, leadership style,

or organizational culture influence individual creativity or innovation performance. In addition, [53] found that self-efficacy has an effect on an individuals' learning and retention capacity and exploration and exploitation activities.

**Hypothesis 2.1.** *Entrepreneurs' self-efficacy positively influences exploration activities.*

**Hypothesis 2.2.** *Entrepreneurs' self-efficacy positively influences exploitation activities.*

In this study, among entrepreneurs' main characteristics, internal control is considered the main explanatory variable for learning activities that mediate entrepreneurial opportunities. Planned behavior theory includes the concept of perceived behavior control [54]. In social psychology, [55] first proposed the concept of the locus of control, which is defined as "people's perception of the source of control over their destiny or actions" [56]. Individuals with an internal locus of control believe that they have control over their lives and those with an external locus of control believe that the control is outside [57]. Chen et al. [58] divide internal controllability and controllability by chance, asserting that we adopt two categories of locus of control with internal control and external control. Internal control refers to a belief in self-control of the events that occur and the consequences of one's actions. With internal control, people believe that their abilities and actions can determine the rewards [59]. As a result, they believe that they can control the outcomes and themselves through their abilities and actions [60]. The internal locus of control involves skills, abilities, personality, and emotions [61]. Most research focuses on the role of the personal traits of the internal locus of control. For example, [62] suggested that the internal locus of control might influence job success. Phares [63] indicated that people with a higher internal locus of control are more likely to easily overcome environmental barriers and more readily learn required skills. Conversely, people with a strong external locus of control are less productive and passive in their work [64]. Past studies have used internal control as an explanatory variable with a direct relationship to entrepreneurial intention [50,65,66]. In this study, it is assumed that the internal locus of control is a trait suitable for ensuring a series of events and that long-term efforts would have a direct impact on all of the various activities for acquiring knowledge related to entrepreneurship. Ajzen [67] asserted that perceived behavior control affects knowledge-sharing activity. In the context of perceived behavior control, an individual's internal control over their own will and ability to impose their will on the environment should have an important influence on the discovery or use of new knowledge in the process of learning and applying new information.

**Hypothesis 3.1.** *Entrepreneurs' internal locus of control positively influences exploration activities.*

**Hypothesis 3.2.** *Entrepreneurs' internal locus of control positively influences exploitation activities.*

In the case of exploration, the discovery and experimentation of new technology is a core activity for firms [5,68]. Exploration focuses on creating variety in experience and thrives on experimentation and free association [69]. According to [70], exploration is a process by which organizations establish experiential variety through experimentation, trialing, and free association. Exploration itself is not efficient for existing business since it focuses on new technologies that are not related to firms' existing knowledge. Exploration is focusing on radically new knowledge by transforming or combining new or existing knowledge [71–74].

In contrast, according to [5], exploitation is the refinement, extension, and enhancement of existing capabilities and technologies. More specifically, exploitation is a process by which organizations create reliability in experience through refinement, production, and focused attention [75]. Exploitation is a process of reutilization to add value to existing knowledge. Such exploitation could increase firms' innovation performance by enhancing competencies in certain knowledge domains that result from specialization [76]. Exploitation normally focuses on refinement and the incremental extension of existing

capabilities [77]; however, exploitation that focuses on innovation changes the existing internal links within a firm. Exploration is a more uncertain and innovative technology or knowledge-investment activity that is not directly related to performance, whereas exploitation is an activity that leverages the knowledge or technology that the company already manages professionally so it can achieve results more rapidly.

**Hypothesis 4.1.** *Entrepreneurs' exploration activities positively influence entrepreneurial intention.*

**Hypothesis 4.2.** *Entrepreneurs' exploitation activities positively influence entrepreneurial intention.*

## 5. Research Methodology

### 5.1. Sampling and Data Collection

A field study was conducted to collect data to test the research model. From 15 February to 8 March 2021, online surveys were distributed to prospective entrepreneurs in 55 universities in metropolitan areas and nationwide in Korea. In order to find prospective entrepreneurs to survey, 55 universities with a start-up curriculum were selected by securing a list of university notifications and universities nationwide. With the help of start-up childcare and start-up education managers in the Seoul metropolitan area and 55 universities nationwide, start-up students were sent e-mails and text messages directly to participate in the online survey. We sent an online link to the questionnaires to 800 potential entrepreneur respondents. We received consent from respondents to participate in the survey and consent to the use of the respondents' personal information within the online questionnaire. In addition, coffee coupons were provided as compensation for the survey response. Of the responses received, eight with duplicate responses or unanswered questions were discarded, yielding 203 usable responses. Frequency analysis was performed using SPSS 20.0 (IBM, USA, Newyork). Among the 203 usable responses, 85 were male (41.9%) and 118 were female (58.1%). The majority of respondents were 20 years old (78.3%). In addition, we asked about the expected industry for entrepreneurship, and the respondents reported science and technology service at 20.2%, followed by accommodation and food service at 13.8% and wholesale and retail at 11.3%. Detailed descriptive statistics of the data related to the respondents are presented in Table 1.

**Table 1.** Descriptive statistics.

| Variables | Items | Frequency | Percent (%) |
|---|---|---|---|
| Gender | Male | 85 | 41.9 |
| | Female | 118 | 58.1 |
| Age | 20s | 159 | 78.3 |
| | 30s | 33 | 16.3 |
| | 40s | 11 | 5.5 |
| Expected industry fields | Agriculture, forestry, and fishing | 8 | 3.9 |
| | Wholesale and retail | 23 | 11.3 |
| | Manufacturing | 20 | 9.9 |
| | Accommodation and catering | 28 | 13.8 |
| | Publishing/video/audio/information services | 13 | 6.4 |
| | Science and technology service | 41 | 20.2 |
| | Professional services (accounting, legal, advertising, consulting, etc.) | 7 | 3.4 |
| | Broadcasting/communication | 5 | 2.5 |
| | Travel/education/rental service | 13 | 6.4 |
| | Medical/health/social welfare | 18 | 8.9 |
| | Culture and arts/sports/leisure | 20 | 9.9 |
| | Etc. | 7 | 3.4 |

*5.2. Questionnaire and Measures*

This research model includes seven constructs with each item gauged by a seven-point Likert scale (1 = strongly disagree, 7 = strongly agree). To ensure construct validity, the items were adapted from previously validated scales. The researchers invited three professionals in the field to review the translated measurement items and research content for validation. A pre-test was conducted to validate the research instrument. Due to low factor loading, 1 item was discarded, leaving 20 items for analysis. The construction concept of this study was referenced in the studies [18,78–80]. The English questionnaire was translated into Korean and used. The content validity was verified by more than five experts in the related field as to whether the meaning of the questionnaire was accurately changed. In addition, prior to the survey, a preliminary survey was conducted on 30 people to increase the ease of answering the survey. Table 2 presents items of constructs.

**Table 2.** Items of constructs.

| Constructs | Items | Reference |
|:---:|:---:|:---:|
| Innovativeness | 2 | |
| Self-efficacy | 4 | [78] |
| Internal locus of control | 3 | |
| Exploration | 3 | [79] |
| Exploitation | 3 | [79] |
| Entrepreneurial intention | 5 | [18,80] |

Innovativeness was assessed using a two-item scale, self-efficacy was measured with a four-item scale, and internal locus of control was assessed based on a three-item scale proposed by [78]. The variables of exploration and exploitation were assessed using a three-item scale, as suggested by [79]. Finally, entrepreneurial intention was measured using a four-item scale [18,80] (See Table 3).

**Table 3.** Measurement.

| Variable | Items | Contents |
|:---:|:---:|:---:|
| Innovativeness | IN1 | I enjoy trying new and unique ways or doing new things. |
| | IN2 | I like new ideas and new things. |
| | IN3 | I'm familiar with the latest trends. |
| Self-efficacy | SE1 | I can start a business with the knowledge I have. |
| | SE2 | I have confidence in every aspect of starting a business. |
| | SE3 | My knowledge is mainly related to entrepreneurship. |
| | SE4 | I am confident that my skills and abilities are not left behind. |
| Internal locus of control | ILC1 | My life is determined by my actions. |
| | ILC2 | I can get what I want if I work hard. |
| | ILC3 | I think I can decide most of what will happen in my life. |
| Exploration | EPR1 | I try to fix problems that I feel are unsatisfactory. |
| | EPR2 | I try to solve unsatisfactory problems well. |
| | EPR3 | I try to introduce new knowledge or methods to solve unsatisfactory problems. |
| Exploitation | EPI1 | I apply (identify, connect, and combine) valuable knowledge I have to start a business or to perform tasks during entrepreneurship education. |
| | EPI2 | I started a business using my existing competencies or performed a task during entrepreneurship education. |

**Table 3.** *Cont.*

| Variable | Items | Contents |
|---|---|---|
| Exploitation | EPI3 | I used my experience accumulated in the past to start a business or solve problems that occur when performing tasks during entrepreneurship education. |
| Entrepreneurial intention | EI1 | I was like that in the past, and I will do anything to start a business now. |
| | EI2 | I was like that in the past, and now I will start a business and put all my efforts into running the business. |
| | EI3 | I was like that in the past, and I decided to start a business in the future. |
| | EI4 | I was and still am, and my professional goal is to become an entrepreneur. |
| | EI5 | I was like that in the past, and I still have high intentions to start a business. |

### 5.3. Common Method Variance and Multicollinearity

This study used the common method variance analysis of Harman's single-factor test [81] and applied the suggestion of [82] about the total variance explained for one common factor, meaning below the cut-off point of 50%. The result explains 32.641%, which confirms that the common method bias in our data set was acceptable. In addition, all constructs' variance-inflation factor (VIF) statistics were examined to assess the multicollinearity problem. As Table 4 shows, all the VIFs were lower than 3.3, indicating that there were no multicollinearity problems. Based on [83], full collinearity VIFs can test common method bias. Since all the VIFs were lower than 3.3, there was no serious common method bias. The multicollinearity problem evaluates the tolerance and VIF that need to be assessed. According to [84]'s recommendations, value tolerance had a threshold of 0.10 and a VIF valued below 10. Table 4 shows that the results of this study meet the requirements.

**Table 4.** Results of multicollinearity testing.

| Model 1 | Unstandardized Coefficients | | Standardized Coefficients | | | Collinearity Statistics | |
|---|---|---|---|---|---|---|---|
| | B | SE | β | t | Sig. | Tolerance | VIF |
| Constant | 0.071 | 0.396 | | 0.180 | 0.858 | | |
| Risk taking | 0.220 | 0.080 | 0.189 | 2.760 | 0.006 | 0.569 | 1.702 |
| Innovativeness | −0.147 | 0.083 | −0.105 | −1.772 | 0.078 | 0.751 | 1.393 |
| Self-efficacy | 0.246 | 0.070 | 0.222 | 3.491 | 0.001 | 0.658 | 1.227 |
| Internal locus of control | 0.180 | 0.094 | 0.119 | 1.916 | 0.057 | 0.692 | 1.324 |
| Exploration | 0.183 | 0.075 | 0.156 | 2.448 | 0.015 | 0.654 | 1.224 |
| Exploitation | 0.288 | 0.058 | 0.306 | 4.986 | 0.000 | 0.707 | 1.224 |

Note: dependent variable: entrepreneurial intention.

## 6. Results

This study adopted partial least-square–structural-equation modeling (PLS–SEM) to analyze the collected data. PLS–SEM was selected for this study's data because PLS–SEM is usually adopted when a data sample size is small and requires testing both formative and reflective measurements or when there is a need to assess the measurement model and structural path coefficients [85,86]. Therefore, it is relatively common to use PLS–SEM for survey data with a small sample size. Based on the above, this study also applied PLS–SEM using SmartPLS 3.0. The analysis contained two steps; the first step included evaluating the measurement model, and the second step involved an assessment of the structural model.

### 6.1. Measurement Model

We tested three measurements to evaluate the internal consistency of the constructs: standardized loadings, composite reliability (CR), and average variance extracted (AVE). As shown in Table 5, reliability analysis testing revealed that the minimal indicator–construct

loadings were higher than 0.70. The CRs for the constructs were higher than 0.80. In addition, all constructs were more significant than the suggested cut-off value of 0.70 [87]. The AVE ranged from 0.559 to 0.720, which is greater than the suggested value of 0.50 [88].

**Table 5.** Reliability and validity estimation.

| Construct | Indicators | Loading | Cronbach's $\alpha$ | CR | AVE |
|---|---|---|---|---|---|
| Innovativeness | IN1 | 0.907 | 0.772 | 0.898 | 0.814 |
| | IN2 | 0.898 | | | |
| Self-efficacy | SE1 | 0.807 | 0.854 | 0.790 | 0.559 |
| | SE2 | 0.867 | | | |
| | SE3 | 0.867 | | | |
| | SE4 | 0.796 | | | |
| Internal locus of control | ILC1 | 0.789 | 0.609 | 0.900 | 0.694 |
| | ILC2 | 0.809 | | | |
| | ILC3 | 0.632 | | | |
| Exploration | EPR1 | 0.793 | 0.784 | 0.874 | 0.697 |
| | EPR2 | 0.848 | | | |
| | EPR3 | 0.863 | | | |
| Exploitation | EPI1 | 0.834 | 0.821 | 0.893 | 0.736 |
| | EPI2 | 0.851 | | | |
| | EPI3 | 0.888 | | | |
| Entrepreneurial intention | EI1 | 0.852 | 0.894 | 0.922 | 0.703 |
| | EI2 | 0.801 | | | |
| | EI3 | 0.884 | | | |
| | EI4 | 0.780 | | | |
| | EI5 | 0.871 | | | |

As Table 6 shows, the discriminant validity of the scales was assessed by comparing the square roots of the AVEs with the correlations among the constructs. If the square roots of AVEs for each construct were greater than their correlations compared to any other construct, it means that the constructs were empirically distinct.

**Table 6.** Analysis of discriminant validity.

| Construct | Innovativeness | Self-Efficacy | Internal Locus of Control | Exploration | Exploitation | Entrepreneurial Intention |
|---|---|---|---|---|---|---|
| Innovativeness | 0.902 | | | | | |
| Self-efficacy | 0.355 | 0.833 | | | | |
| Internal locus of control | 0.329 | 0.162 | 0.747 | | | |
| Exploration | 0.445 | 0.277 | 0.402 | 0.835 | | |
| Exploitation | 0.430 | 0.437 | 0.346 | 0.428 | 0.858 | |
| Entrepreneurial intention | 0.460 | 0.555 | 0.192 | 0.397 | 0.528 | 0.839 |

### 6.2. Structural Model Assessment

Because the evaluation of the measurement model confirmed the validity and reliability issue, the structural model needed to evaluate the hypotheses [30]. The path coefficients of the construct were measured through SmartPLS 3.0 analysis. In addition, using the bootstrapping procedure (5000 re-samples), the significance of the paths of the structural model was examined. Analysis of the structural model indicates that the data fit the index (SRMR = 0.07, d_ULS = 1.355, d_G = 0.534, $\chi^2$ = 653.793, NFI = 0.731, RMS θ = 0.171). The results are presented in Figure 2.

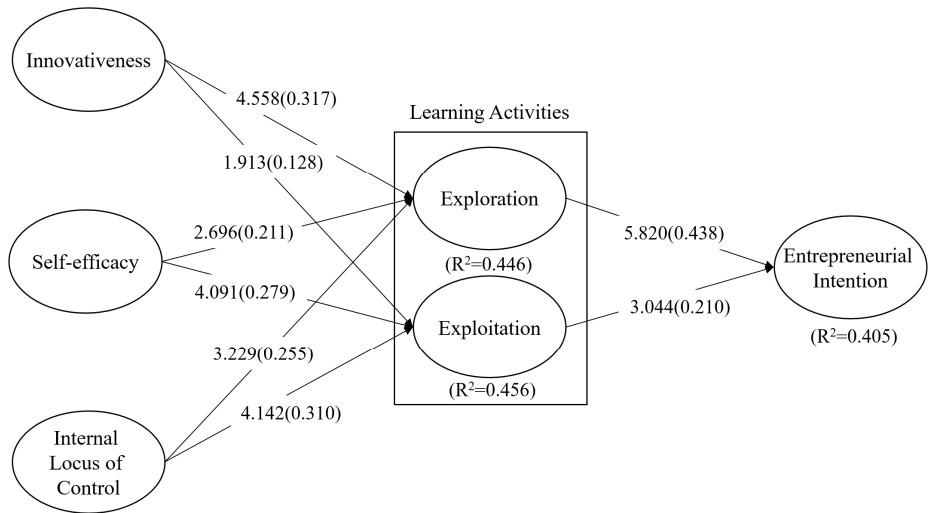

**Figure 2.** Results of the research model.

As Table 7 shows, the test results indicate that all hypotheses are supported. Exploration was significantly influenced by innovativeness (β = 0.075 t = 4.145, $p < 0.05$), self-efficacy (β = 0.068 t = 1.892, $p < 0.058$), and internal locus of control (β = 0.068 t = 4.127, $p < 0.05$), suggesting support for H1.1, H2.1, and H3.1. However, there was a lack of effect size on self-efficacy in exploration, so we concluded that the H2.1 is rejected. We can assume that even if the prospective entrepreneurs possess self-efficacy about new start-ups, exploration activities naturally do not confirm the outcomes. Thus, self-efficacy could not strongly relate to the exploration activities that require uncertainties and accidental findings. For exploitation activities, innovativeness (β = 0.080 t = 3.180, $p < 0.05$), self-efficacy (β = 0.070 t = 4.555, $p < 0.05$), and internal locus of control (β = 0.078 t = 2.707, $p < 0.05$) positively influenced exploitation, suggesting support for H1.2, H2.2, and H3.2. Finally, as mediators between personal traits and entrepreneurial intention, exploration (β = 0.068 t = 3.062, $p < 0.05$) and exploitation (β = 0.074 t = 5.929, $p < 0.05$) significantly influenced entrepreneurial intention, suggesting support for H4.1 and H4.2.

**Table 7.** Results of the hypothesis tests.

| Hypotheses | Effect Size | T-Statistics | *p*-Value | Hypothesis Supported |
|---|---|---|---|---|
| Hypothesis 1.1. Innovativeness→Exploration activities | 0.144 | 4.218 | 0.000 | Supported |
| Hypothesis 1.2. Innovativeness→Exploitation activities | 0.090 | 3.144 | 0.002 | Supported |
| Hypothesis 2.1. Self-efficacy→Exploration activities | 0.018 | 1.897 | 0.058 | Not supported |
| Hypothesis 2.2. Self-efficacy→Exploitation activities | 0.182 | 4.562 | 0.000 | Supported |
| Hypothesis 3.1. Internal locus of control→Exploration activities | 0.202 | 4.149 | 0.000 | Supported |
| Hypothesis 3.2. Internal locus of control→Exploitation activities | 0.111 | 2.664 | 0.008 | Supported |
| Hypothesis 4.1. Exploration activities→Entrepreneurial intention | 0.050 | 3.052 | 0.002 | Supported |
| Hypothesis 4.2. Exploitation activities→Entrepreneurial intention | 0.307 | 5.809 | 0.000 | Supported |

### 7. Conclusions

This study sought to investigate the effect of entrepreneurs' individual characteristics on entrepreneurial intention, assessing individuals' learning activities as moderators. The unique characteristics of individuals alone cannot explain the impact on entrepreneurial intention, which requires the long-term resolution of various emerging challenges and a long-term perspective. Therefore, it was intended to understand the mechanisms that affect entrepreneurial intention using the nature of the individual's learning activity as a mediator. In this vein, the results of this research successfully support our research questions about the hidden mechanisms between personal traits and entrepreneurial intention. In this regard, this study offers several theoretical and practical implications.

For theoretical implications, first, the research on the factors that affect entrepreneurial intention contributes to expanding and deepening the research on entrepreneurial intention, as it identified and validated major factors and individual characteristics that could affect entrepreneurial intention. Entrepreneurs' innovativeness, self-efficacy, and internal control can be seen as the individual characteristics of entrepreneurs that are necessary to solve various problems under the uncertainty of starting a business. Although many studies have studied the effect of individual characteristics on entrepreneurship intention, this study judged that entrepreneurs' individual characteristic variables will explore new knowledge and skills and affect exploitation activities to solve problems arising from entrepreneurship. As a result, it was found that entrepreneurs' innovation, self-efficacy, and internal control are associated with exploration and exploitation activities. The results of this study suggest that entrepreneurship intention can be related to various skills and abilities of individuals, not only including their personal traits but also the characteristics that can be developed by them. Second, from the perspective of the theory of organizational learning, most previous studies judged learning activities as investments in new technologies or knowledge within a firm or the expansion of knowledge. This study is of theoretical significance, as it applied the activities presented in the organizational learning theory to prospective entrepreneurs' activities in the start-up field. Third, in this study, the activities for ambidextrous learning for acquiring new knowledge and skills for future start-up companies and to improve, expand, and improve existing capabilities and skills being related to the entrepreneurship intention was confirmed. The prospective entrepreneurs were found to have an effect on the performance of entrepreneurship intention while maintaining a balance between exploration activities and utilization activities in order to establish corporate growth and internal stability.

The practical implications include some important insights. To increase the intention of starting a business, education or training development was not possible for the characteristics and temperament of prospective entrepreneurs based on previous research; however, the results of the hypotheses presented in this study are significant, as they were demonstrated to be sufficiently strengthened by the nature of the learning activities engaged in or through the support of the relevant institutions. Therefore, it is meaningful to strengthen training and learning activities that can develop and reinforce the individual characteristics identified to directly enhance start-up intentions by facilitating start-up-related problem-solving activities.

The limitations of this study and future research directions are presented as follows. First, in this study, research was conducted only for prospective entrepreneurs. The characteristics of entrepreneurs according to the start-up period will have different effects on learning activities, so it is necessary to subdivide entrepreneurs and examine how their personal characteristics affect learning activities. Second, since exploration and exploitation activities, which are variables of learning activities, have different performance goals in terms of content and time, they should be measured as different dependent variables in empirical studies. However, in this study, since start-up intention was used as a dependent variable, a study examining the relationship with various dependent variables will be needed in the future. As for the limitations of this study, the research model can be developed along with other variables in the future for the causal relationship, which

is not clearly elucidated, along with other personality characteristics of the prospective founders not covered by this study. In addition, given that this study conducted a survey on prospective entrepreneurs, it is clear how learning activities will have a direct effect on the performance of start-up and, through this, will have a positive effect on actually starting a start-up. However, in future research, it is necessary to conduct research targeting those who are currently engaged in start-ups.

**Author Contributions:** Conceptualization, S.L. and B.-K.K.; Data curation, M.-J.K.; Formal analysis, M.-J.K.; Funding acquisition, B.-K.K.; Investigation, M.-J.K.; Methodology, S.L.; Project administration, B.-K.K.; Resources, B.-K.K.; Validation, B.-K.K.; Writing—original draft, S.L. and M.-J.K.; Writing—review & editing, S.L. and B.-K.K. All authors have read and agreed to the published version of the manuscript.

**Funding:** This research was funded by the Ministry of Education of the Republic of Korea and the National Research Foundation of Korea [NRF-2019S1A5C2A02082342].

**Institutional Review Board Statement:** Not applicable.

**Informed Consent Statement:** Not applicable.

**Data Availability Statement:** The data presented in this study are available on request from the corresponding author.

**Conflicts of Interest:** The authors declare no conflict of interest.

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
