# Peer review of "Factors Influencing Entrepreneurial Intention: Focusing on Individuals’ Knowledge Exploration and Exploitation Activities"

_2199-8531, doi:10.3390/joitmc8030165_

Round 1

Reviewer 1 Report

This study investigates the factors that affect entrepreneurial intention by focusing on individuals' knowledge exploration and exploitation activities. The authors collected 203 usable data from prospective entrepreneurs (i.e., university students) in metropolitan areas nationwide in Korea and adopted partial least squares-structural equation modeling to analyze them. By doing so, the authors argued that they found the moderating role of individuals' learning activities (i.e., exploration and exploitation) over the individuals' characteristics (i.e., innovativeness, self-efficacy, and internal locus of control) and the entrepreneurial intention. This might be an interesting manuscript as the authors explore the possibility of bridging the individual characteristics and the intention by applying the organizational learning theory to the personal level of learning.

While this is an interesting and solid study, several points require more clarification and/or careful examination. The authors could strengthen their manuscript by addressing the following concerns.

1) L292-293: The authors sent an online questionnaire link to 800 potential entrepreneur respondents. But how did the authors select 800 respondents? The way of screening 800 respondents should clarify this to avoid potential sampling bias.

2) L313-314: Please describe the pre-test procedure.

3) Section 5.2: Questionaire items administered should be shown in the body of the text or appendix.

4) L365-L366: In figure 1, the significant path, path coefficients, p-value results, and variance of the endogenous constructs are not shown, although the authors mentioned so.

5) L368-L375 & Table 6: 

5-1) Are beta (β) path coefficients? Then, what are the values of coefficients in Table 6? I believe both should be the same values.

5-2) If path coefficients (β) are correctly shown (below 0.10), the values seem too low to be seen as supporting the hypothesized effects without the inspection of the effect size, even with the small p-values. I strongly recommend calculating and showing Cohen's f-squared effect size coefficients for all path coefficients. With the effect size, the authors can ascertain if the effects indicated by path coefficients are small, medium, or large.

5-3) The p-value of self-efficacy is larger than p = .05. Should the hypothesis be rejected?

6) L401-L407: The authors argued the appearance of ambidextrous organizations. Also, they described the characteristics of the actual entrepreneurs. However, this seems an overstatement. The authors collected data only from prospective entrepreneurs, not the actual entrepreneurs: therefore, the discussion/conclusion should remain within the scope regarding prospective entrepreneurs.

7) L408-416: The authors argue that the personal traits are enhanced/strengthened by the nature of learning activities engaged or through the support of the relevant institution. However, in the present study, the hypothesis test only considers the path from personal traits to learning activities. This paper does not analyze and discuss the reverse direction/path (from learning activities to personal traits). Thus, this seems an overstatement. The practical contribution must be extracted from the results analyzed.

8) The authors must show ethical consideration in this study since this is human research. 

To increase the readability, I recommend the following points.

9) L37: What kind of experience? Please specify.

10) L40 - L54: Although the rationale behind applying organizational learning theory to the individual level can be seen in the Literature Review section, you can insert a few sentences to explain the rationale.

11) Section 2.1. Entrepreneurial Intention: Why did the authors adopt only three personal traits (innovativeness, self-efficacy, and locus of control) among other potential factors introduced in the section? Please explain.

12) L163: Insert the parenthesis to the references. (Before Eisenhardt, et al. 2010).

13) L308: Correct the position of parentheses of references. From (Koh, 1996) to Koh (1996).

Author Response

Executive Summary of Modifications to JOItmC-1884397

We are grateful to the Editor and the three reviewers for their time and efforts to improve this manuscript. We have incorporated in this revision all the suggestions made by the review team. This document contains our point-by-point responses to all comments. The major improvements in this revision are summarized below:

  1. Literature Review: As all review panelists suggested, we have thoroughly reviewed the recent literature on entrepreneurship intention (Section 2.1) and organizational learning theory (Section 2.2).

  1. Theoretical Contributions and Conclusions: Based on Reviewer’s advice, we have significantly improved the Theoretical Contributions and Conclusion sections by excluding overstated parts.

  1. Methods and Results: Based on Reviewer 1, we have revised the Table 6 and add additional explanation of effects of each relationships.

  1. We have also made numerous other changes to address the concerns expressed in the review packet, including the issues related to updating current research and the contribution of our study.

Once again, we thank the Editor and the three reviewers so much for their support and guidance. We are sincerely indebted to all members of the editorial team for their help in creating a paper of much higher quality. We hope that our revised manuscript has satisfactorily addressed all concerns raised in the last round of review. Nevertheless, we will be glad to make any further changes you deem necessary.

With warm regards,

The authors

Reviewer 1

Comments

Our Responses

This study investigates the factors that affect entrepreneurial intention by focusing on individuals' knowledge exploration and exploitation activities. The authors collected 203 usable data from prospective entrepreneurs (i.e., university students) in metropolitan areas nationwide in Korea and adopted partial least squares-structural equation modeling to analyze them. By doing so, the authors argued that they found the moderating role of individuals' learning activities (i.e., exploration and exploitation) over the individuals' characteristics (i.e., innovativeness, self-efficacy, and internal locus of control) and the entrepreneurial intention. This might be an interesting manuscript as the authors explore the possibility of bridging the individual characteristics and the intention by applying the organizational learning theory to the personal level of learning.

While this is an interesting and solid study, several points require more clarification and/or careful examination. The authors could strengthen their manuscript by addressing the following concerns.

Thanks for serving as the Editor of our paper. We appreciate the high-quality feedback from the two experts chosen as part of this review team. We have done our best to thoroughly address your concerns and hope that the revision addresses the concerns raised, including the contribution issue, to your satisfaction.

The authors sent an online questionnaire link to 800 potential entrepreneur respondents. But how did the authors select 800 respondents? The way of screening 800 respondents should clarify this to avoid potential sampling bias.

Thank you. The process of selecting 800 survey subjects in this study is explained in more detail as follows to make it easier for readers to understand.

“In order to find prospective entrepreneurs who are surveyed, 55 universities with start-up curriculum were selected by securing a list of university notifications and universities nationwide. With the help of start-up childcare and start-up education managers in the Seoul metropolitan area and 55 universities nationwide, start-up students were sent e-mails and text messages directly to participate in the online survey.”

Please see p. 10 in manuscript

Please describe the pre-test procedure.

Thank you for your kind comment. To help readers understand, the pre-test process is expressed in more detail as follows.

“The construction concept of this study was referenced in the studies of Koh (1996), Prieto et al. (2009), Crant (1996), Diaz-Garcia and Jimenez-Moreno (2010). The English questionnaire was translated into Korean and used. The content validity was verified by more than five experts in the related field as to whether the meaning of the questionnaire was accurately changed. In addition, prior to the survey, a preliminary survey was conducted on 30 people to increase the ease of answering the survey.”

Please see p. 11 in manuscript

Questionaire items administered should be shown in the body of the text or appendix.

Thank you for the reviewer’s comment. As you ,mentioned, we filled out and presented the questionnaire in Table 2 as follows.

Please see p. 12 in manuscript

In figure 1, the significant path, path coefficients, p-value results, and variance of the endogenous constructs are not shown, although the authors mentioned so.

Thank you for the reviewer’s comment. In order for readers to better understand the results of this study, the indicators mentioned in Figure 2 is presented so that they can be displayed.

Please see p. 16 in manuscript

5-1) Are beta (β) path coefficients? Then, what are the values of coefficients in Table 6? I believe both should be the same values

Thank you for the reviewer’s comment. As you said, effect size is presented for all variables. The effect size is not high, but it is so small that it is difficult to support the hypothesis only for self-efficiency and expansion, so the advantage is specified in the result analysis part (See Table 6).

Please see p. 15 in manuscript.

5-2) If path coefficients (β) are correctly shown (below 0.10), the values seem too low to be seen as supporting the hypothesized effects without the inspection of the effect size, even with the small p-values. I strongly recommend calculating and showing Cohen's f-squared effect size coefficients for all path coefficients. With the effect size, the authors can ascertain if the effects indicated by path coefficients are small, medium, or large.

Thank you for the reviewer’s comment. In this study, it is assumed that the Path loading value is statistically supported even if the P value criterion is less than 1. However, since the effect size is very small, the path loading value in this study is concluded to reject.

Please see p. 16-17 in manuscript

5-3) The p-value of self-efficacy is larger than p = .05. Should the hypothesis be rejected?

Thank you for your comment. In this study, regarding the effect of self-efficacy on exploratory activities, the p-valeu was .058 to a value greater than .05, and the result was rejected, as shown below in Table 6. In addition, in the results section, the following explanations were added for the causes of these statistical results.

“However, there is a lack of effect size on self-efficacy in exploration, we conclude that the H 2-1 is rejected. We can assume that even the prospective entrepreneur possess self-efficacy about new start-up, exploration activities naturally do not confirm the outcomes. Thus, self-efficacy could not strongly related to the exploration activities which require the uncertainties and accidental findings.”

Please see p. 16 in manuscript

The authors argued the appearance of ambidextrous organizations. Also, they described the characteristics of the actual entrepreneurs. However, this seems an overstatement. The authors collected data only from prospective entrepreneurs, not the actual entrepreneurs: therefore, the discussion/conclusion should remain within the scope regarding prospective entrepreneurs.

Thank you for the reviewer’s comment. As you said, we will delete the overstatement part of the part that was not analyzed in this study. Rather, we will focus on the fact that personal trait can affect learning activities themselves, and that personal characteristics such as innovative, self-efficacy, or locus of control are related to learning activities that are important for solving problems.

“Third, in this study, the activities for ambidextrous learning for acquiring new knowledge and skills for future start-up companies and to improve, expand, and improve existing capabilities and skills is related to the entrepreneurship intention was confirmed. The prospective entrepreneurs were found to have an effect on the performance of entrepreneurship intention while maintaining a balance between exploration activities and utilization activities in order to establish corporate growth and internal stability.”

Please see p. 18 in manuscript

: The authors argue that the personal traits are enhanced/strengthened by the nature of learning activities engaged or through the support of the relevant institution. However, in the present study, the hypothesis test only considers the path from personal traits to learning activities. This paper does not analyze and discuss the reverse direction/path (from learning activities to personal traits). Thus, this seems an overstatement. The practical contribution must be extracted from the results analyzed.

Thank you for the reviewer’s comment. As you said, we will delete the overstatement part of the part that was not analyzed in this study. Rather, we will focus on the fact that personal trait can affect learning activities themselves, and that personal characteristics such as innovative, self-efficacy, or locus of control are related to learning activities that are important for solving problems.

“The results of this study suggest that the entrepreneurship intention can be related to various skills and abilities of individuals not only including their personal traits but also the characteristics that can be developed by themselves.”

Please see p. 17-18 in manuscript

The authors must show ethical consideration in this study since this is human research. 

Following your comment, we add the specific procedures of studies. At the beginning of the survey, this study confirm the purpose of the study using information, and clarified the contents of the survey participation reward. It has been described as follows.

“We received consent from respondents to participate in the survey and consent to the use of respondents' personal information within the online questionnaire. In addition, coffee coupons were provided as compensation for the survey response.”

Please see p. 10 in manuscript

To increase the readability, I recommend the following points.

9) L37: What kind of experience? Please specify.

Thank you for the reviewer’s comment. we explained additionally as below.

“Existing research suggests that learning and direct and indirect experience about participating of starting start-up can affect start-up initiatives (Gist, 1987).”

Please see p. 2 in manuscript

10) L40 - L54: Although the rationale behind applying organizational learning theory to the individual level can be seen in the Literature Review section, you can insert a few sentences to explain the rationale.

Thank you for the reviewer’s comment. The reason why the organizational learning theory was applied to the individual was additionally explained as follows.

“Recently, a few study start to emphasize that individual also could engage the routine of learning activities and that results in the organizational performance (Eisenhardt et al., 2010; Lee & Meyer-Doyle, 2017).”

Please see p. 2 in manuscript

11) Section 2.1. Entrepreneurial Intention: Why did the authors adopt only three personal traits (innovativeness, self-efficacy, and locus of control) among other potential factors introduced in the section? Please explain.

Thank you for the reviewer’s comment. In your comments and in this study, three main variables affect learning activities. There are so many variables that can also belong to individual characteristics. The reason for using only those three variables was added as follows.

“In this study, a study was conducted for prospective entrepreneurs who should perform entrepreneurship based on innovative ideas. In this study, innovative output was assumed to be the starting start-up, and the research model was presented by limiting the research model to innovativeness, self-efficacy, and internal locus of control by focusing on individual characteristics that can affect the learning activities.”

Please see p. 7 in manuscript

12) L163: Insert the parenthesis to the references. (Before Eisenhardt, et al. 2010).

Thanks for the reviewer’s comment. We revised it.

Please see p. 6 in manuscript

13) L308: Correct the position of parentheses of references. From (Koh, 1996) to Koh (1996).

Thanks for the reviewer’s comment. We revised it.

Please see p. 11 in manuscript

Table 3 Measurement

Variable

Items

Contents

Innovativeness

IN1

I enjoy trying new and unique ways or doing new things

IN2

I like new ideas and new things

IN3

I'm familiar with the latest trends

Self-efficacy

SE1

I can start a business with the knowledge I have

SE2

I have confidence in every aspect of starting a business.

SE3

My knowledge is mainly related to entrepreneurship.

SE4

I am confident that my skills and abilities are not left behind

Internal locus of control

ILC1

My life is determined by my actions

ILC2

I can get what I want if I work hard

ILC3

I think I can decide most of what will happen in my life

Exploration

EPR1

I try to fix problems that I feel are unsatisfactory

EPR2

I try to solve unsatisfactory problems well

EPR3

I try to introduce new knowledge or methods to solve unsatisfactory problems

Exploitation

EPI1

I apply (identify, connect, and combine) valuable knowledge I have to start a business or to perform tasks during entrepreneurship education.

EPI2

I started a business using my existing competencies or performed a task during entrepreneurship education.

EPI3

I used my experience accumulated in the past to start a business or solve problems that occur when performing tasks during entrepreneurship education.

Entrepreneurial

intention

EI1

I was like that in the past and I will do anything to start a business now.

EI2

I was like that in the past, and now I will start a business and put all my efforts into running the business.

EI3

I was like that in the past, and I decided to start a business in the future.

EI4

I was and still am, my professional goal is to become an entrepreneur.

EI5

I was like that in the past, and I still have high intentions to start a business.

Figure 2. Results of Research Model

Table 6 Results of Hypotheses Test

Hypotheses

Effect size

T Statistics

p-value

Hypothesis Supported

Hypothesis 1-1. Innovativeness → Exploration activities

.144

4.218

.000

Supported

Hypothesis 1-2. Innovativeness → Exploitation activities

.090

3.144

.002

Supported

Hypothesis 2-1. Self-efficacy → Exploration activities

.018

1.897

.058

Supported

Hypothesis 2-2. Self-efficacy → Exploitation activities

.182

4.562

.000

Supported

Hypothesis 3-1. Internal locus of control → Exploration activities

.202

4.149

.000

Supported

Hypothesis 3-2. Internal locus of control → Exploitation activities

.111

2.664

.008

Supported

Hypothesis 4-1. Exploration activities → Entrepreneurial intention

.050

3.052

.002

Supported

Hypothesis 4-2. Exploitation activities → Entrepreneurial intention

.307

5.809

.000

Supported

Dependent variable: Entrepreneurial Intention; *p < 0.1; **p < 0.05; ***p < 0.0

Reviewer 2 Report

This paper is related to an interesting topic, namely Factors influencing entrepreneurial intention: Focusing on individuals’ knowledge exploration and exploitation activities for JOItmC

However, the manuscript needs to be further developed in order to meet the expected academic requirements.

In Introduction Provide a stronger motivation for the study
The authors need to perform a detailed literature review.
Develop the model in a systematic manner using all the appropriate literature (review the previous theories and models)

The theoretical development for this present study is weak. The theories in the field should be discussed in relation to entrepreneurial intention.

Implications section is missing.

Please add

1.      Theoretical implications

2.      Practical implications

How general are your results? These have to be of interest to the whole community. Relate these with your limitations.

     What are the limitations of your work?

Answer your research question in the conclusions; what did we learn compared with current, significant research (up to 2021).

Author Response

Executive Summary of Modifications to JOItmC-1884397

We are grateful to the Editor and the three reviewers for their time and efforts to improve this manuscript. We have incorporated in this revision all the suggestions made by the review team. This document contains our point-by-point responses to all comments. The major improvements in this revision are summarized below:

  1. Literature Review: As all review panelists suggested, we have thoroughly reviewed the recent literature on entrepreneurship intention (Section 2.1) and organizational learning theory (Section 2.2).

  1. Theoretical Contributions and Conclusions: Based on Reviewer’s advice, we have significantly improved the Theoretical Contributions and Conclusion sections by excluding overstated parts.

  1. Methods and Results: Based on Reviewer 1, we have revised the Table 6 and add additional explanation of effects of each relationships.

  1. We have also made numerous other changes to address the concerns expressed in the review packet, including the issues related to updating current research and the contribution of our study.

Once again, we thank the Editor and the three reviewers so much for their support and guidance. We are sincerely indebted to all members of the editorial team for their help in creating a paper of much higher quality. We hope that our revised manuscript has satisfactorily addressed all concerns raised in the last round of review. Nevertheless, we will be glad to make any further changes you deem necessary.

With warm regards,

The authors

Reviewer 2

Comments

Our Response

In Introduction Provide a stronger motivation for the study

Thank you The research purpose of this study has been added to the Introduction as follows.

“In this study, we assume that the relationship between personal traits and entrepreneurship intention are not directly associated with, but the relationship can be mediated by the ability to recognize various environments and situations and obtain and utilize information appropriate to the situation. This research try to examine the role of learning activities.”

Please see p. 1 in manuscript

The authors need to perform a detailed literature review.
Develop the model in a systematic manner using all the appropriate literature (review the previous theories and models)

Thank you for your comment. We updated the literature review section with latest studies as follows.

“Recently. entrepreneurial intention-related research starts to investigate the factors that mediate personal traits to entrepreneurial intention. With the same vein, we tried to find factors as mediators. For example, Awwad & Al-Aseer (2021) examined how entrepreneurial alertness mediates when big five personality traits influence entrepreneurial intention. Gill et al. (2021) also empirically investigated whether entrepreneurial self-efficacy mediates the influence when cognitive flexibility and entrepreneurial awareness affect entrepreneurial intention. In addition, Tantawy et al., (2021) examined the mediating role of attitude and creative process engagement in the relationship between creative self-efficacy and entrepreneurial intention. Although many recent studies have only looked at the direct path of personal traits that affect entrepreneurial intention, recently, even if they have the same personal trait, the studies start to focus on mediators which is unique abilities of entrepreneurs or prospective entrepreneurs.”

Please see p. 5 in manuscript

“Previous studies related to entrepreneurship have examined on how entrepreneurship orientation is related to organizational learning within companies (Shah et al., 2021). The number of studies has been limited to organizational-level studies, suggesting that the corporate atmosphere and environment have a positive effect on the organizational learning process at the firm level, which in turn has a positive effect on innovation performance. On the contrary, this study differs from previous studies in that it studied exploration and exploitation activities, which are individual activities that affect individual entrepreneurial intention.”

[Reference]

Shah, S. A., Shah, M. H., Matloob, S., & Abbas, S. I. (2021). Managing Organizational performance of pharmaceutical SMEs in Indonesia: Examining the Role of Entrepreneurial Orientation and Organizational Learning in a Mediated Model. Annals of Contemporary Developments in Management & HR (ACDMHR), 3(1), 24-32.

Please see p. 6 in manuscript

The theoretical development for this present study is weak. The theories in the field should be discussed in relation to entrepreneurial intention

As you suggested, we have provided an additional explanation of the mediating role to expand the discussion and theoretical contribution to entrepreneurial intention.

“Recently. entrepreneurial intention-related research starts to investigate the factors that mediate personal traits to entrepreneurial intention. With the same vein, we tried to find factors as mediators. For example, Awwad & Al-Aseer (2021) examined how entrepreneurial alertness mediates when big five personality traits influence entrepreneurial intention. Gill et al. (2021) also empirically investigated whether entrepreneurial self-efficacy mediates the influence when cognitive flexibility and entrepreneurial awareness affect entrepreneurial intention. In addition, Tantawy et al., (2021) examined the mediating role of attitude and creative process engagement in the relationship between creative self-efficacy and entrepreneurial intention. Although many recent studies have only looked at the direct path of personal traits that affect entrepreneurial intention, recently, even if they have the same personal trait, the studies start to focus on mediators which is unique abilities of entrepreneurs or prospective entrepreneurs.”

[Reference]

Awwad, M. S., & Al-Aseer, R. M. N. (2021). Big five personality traits impact on entrepreneurial intention: the mediating role of entrepreneurial alertness. Asia Pacific Journal of Innovation and Entrepreneurship.

Gill, S. A., Bencheva, N., Karayel, S., & Usman, M. (2021). Does entrepreneurial self-efficacy moderate effects of cognitive flexibility and entrepreneurial alertness on entrepreneurial intentions?. Entrepreneurial Business and Economics Review, 9(3), 25-41.

Tantawy, M., Herbert, K., McNally, J. J., Mengel, T., Piperopoulos, P., & Foord, D. (2021). Bringing creativity back to entrepreneurship education: Creative self-efficacy, creative process engagement, and entrepreneurial intentions. Journal of Business Venturing Insights, 15, e00239.

Please see p. 5 in manuscript

Implications section is missing. Please add 1.      Theoretical implications, 2.      Practical implications

Thank you. The theoretical and practical implications of this study are presented below.

“For theoretical implication, first, the research on the factors that affect entrepreneurial intention has contributed to expanding and deepening the research on entrepreneurial intention as it identified and validated major factors and individual characteristics that could affect entrepreneurial intention. Entrepreneur's innovativeness, self-efficacy, and internal control can be seen as the individual characteristics of entrepreneurs necessary to solve various problems under the uncertainty of starting a business. While many studies have studied the effect of individual characteristics on entrepreneurship intention, this study judged that entrepreneurs' individual characteristic variables will explore new knowledge and skills and affect exploitation activities to solve problems arising from entrepreneurship. As a result, it was found that entrepreneurs' innovation, self-efficacy, and internal control are associated with exploration and exploitation activities. The results of this study suggest that the entrepreneurship intention can be related to various skills and abilities of individuals not only including their personal traits but also the characteristics that can be developed by themselves. Second, from the perspective of the theory of organizational learning, most previous studies judge learning activities as investments in new technologies or knowledge within a firm or the expansion of knowledge. This study is of theoretical significance as it applied the activities presented in the organizational learning theory to prospective entrepreneurs’ activities in the start-up field.  Third, in this study, the activities for ambidextrous learning for acquiring new knowledge and skills for future start-up companies and to improve, expand, and improve existing capabilities and skills is related to the entrepreneurship intention was confirmed. The prospective entrepreneurs were found to have an effect on the performance of entrepreneurship intention while maintaining a balance between exploration activities and utilization activities in order to establish corporate growth and internal stability.

Practical implication include some important insights. To increase the intention of starting a business, education or training development was not possible for the characteristics and temperament of prospective entrepreneurs based on previous research; however, the results of the hypotheses presented in this study are significant as they are demonstrated to be sufficiently strengthened by the nature of the learning activities engaged or through the support of the relevant institutions. Therefore, it is meaningful to strengthen training and learning activities that can develop and reinforce the individual characteristics identified to directly enhance start-up intentions by facilitating start-up-related problem-solving activities.”

Please see p. 17-18 in manuscript

    What are the limitations of your work?

How general are your results? These have to be of interest to the whole community. Relate these with your limitations.

Thank you. The limitations of this study have been added as follows.

“As for the limitations of this study, the research model can be developed along with other variables in the future for the causal relationship that is not clearly elucidated along with other personality characteristics of the prospective founders not covered by this study. In addition, given that this study conducted a survey on prospective entrepreneurs, it is clear how learning activities will have a direct effect on the performance of start-up and, through this, will have a positive effect on actually starting a start-up. However, in future research, it is necessary to conduct research targeting those who are currently engaged in start-up.”

Please see p. 18 in manuscript

Answer your research question in the conclusions; what did we learn compared with current, significant research (up to 2021).

Thank you. This study describes how the purpose of this study was achieved in the results section, and furthermore, based on recent studies in the literature study, the direction of the development of the relevant research field is also added as below.

“In this vein, the results of this research successfully supports our research questions about the hidden mechanisms between personal traits and entrepreneurial intention. In this regard, this study offers several theoretical and practical implication.”

Please see p. 17 in manuscript

“Recently. entrepreneurial intention-related research starts to investigate the factors that mediate personal traits to entrepreneurial intention. With the same vein, we tried to find factors as mediators. For example, Awwad & Al-Aseer (2021) examined how entrepreneurial alertness mediates when big five personality traits influence entrepreneurial intention. Gill et al. (2021) also empirically investigated whether entrepreneurial self-efficacy mediates the influence when cognitive flexibility and entrepreneurial awareness affect entrepreneurial intention. In addition, Tantawy et al., (2021) examined the mediating role of attitude and creative process engagement in the relationship between creative self-efficacy and entrepreneurial intention. Although many recent studies have only looked at the direct path of personal traits that affect entrepreneurial intention, recently, even if they have the same personal trait, the studies start to focus on mediators which is unique abilities of entrepreneurs or prospective entrepreneurs.”

Please see p. 5 in manuscript

Round 2

Reviewer 2 Report

please add the implications in separate section

1.      Theoretical implications

2.      Practical implications

Author Response

Thank you very much for your kind suggestion. However, we don't understand that  add the implications in separate section. I would be very grateful if you could explain your suggestion in detail.